# Extracellular ATP: A Feasible Target for Cancer Therapy

**DOI:** 10.3390/cells9112496

**Published:** 2020-11-17

**Authors:** Valentina Vultaggio-Poma, Alba Clara Sarti, Francesco Di Virgilio

**Affiliations:** Department of Medical Sciences, University of Ferrara, via Luigi Borsari 46, 44121 Ferrara, Italy; vltvnt@unife.it (V.V.-P.); srtlcl@unife.it (A.C.S.)

**Keywords:** extracellular ATP, cancer, tumor microenvironment, purinergic signaling

## Abstract

Adenosine triphosphate (ATP) is one of the main biochemical components of the tumor microenvironment (TME), where it can promote tumor progression or tumor suppression depending on its concentration and on the specific ecto-nucleotidases and receptors expressed by immune and cancer cells. ATP can be released from cells via both specific and nonspecific pathways. A non-regulated release occurs from dying and damaged cells, whereas active release involves exocytotic granules, plasma membrane-derived microvesicles, specific ATP-binding cassette (ABC) transporters and membrane channels (connexin hemichannels, pannexin 1 (PANX1), calcium homeostasis modulator 1 (CALHM1), volume-regulated anion channels (VRACs) and maxi-anion channels (MACs)). Extracellular ATP acts at P2 purinergic receptors, among which P2X7R is a key mediator of the final ATP-dependent biological effects. Over the years, P2 receptor- or ecto-nucleotidase-targeting for cancer therapy has been proposed and actively investigated, while comparatively fewer studies have explored the suitability of TME ATP as a target. In this review, we briefly summarize the available evidence suggesting that TME ATP has a central role in determining tumor fate and is, therefore, a suitable target for cancer therapy.

## 1. Introduction

Our understanding of the physiological role of ATP has been greatly expanded ever since its first isolation from skeletal muscle [1]. It is now clear that, besides energy transduction, ATP plays a fundamental role in signaling, as it serves as a phosphate-group donor for substrate activation in metabolic reactions, is required for the biosynthesis of the intracellular second messenger cyclic adenosine monophosphate (cAMP) and mediates intercellular communication as a bona fide extracellular messenger [2]. In 1970, Geoff Burnstock demonstrated that extracellular ATP (eATP) is the transmitter substance released by non-adrenergic inhibitory nerves [3], and later in 1972 formulated his “purinergic hypothesis” postulating that ATP is released by most cells as an extracellular signaling molecule [4]. The definitive sanction of the purinergic hypothesis was provided by cloning of the first P2Y metabotropic [5] and the first P2X1 ionotropic receptors [6], soon followed by all the other members of the P2Y and P2X sub-families.

Although early studies focused on the role of purinergic receptors in neurotransmission, as early as 1980 it was suggested that specific plasma membrane receptors for extracellular ATP were also expressed by inflammatory and cancer cells [7,8]. Today there is a wide consensus that eATP and other nucleotides, and their plasma membrane receptors play a central role in tumor cell proliferation and immune cell regulation [9,10].

ATP possesses all the features of an ideal extracellular messenger: (a) is virtually absent in the extracellular space under physiological conditions (estimated concentration 10–100 nmol/L); (b) is stored in very high amounts within the cells (from 5 to 10 mmol/L); (c) is water-soluble and freely diffusible in the extracellular space due to negatively charged phosphate residues; (d) is rapidly degraded by ubiquitous extracellular nucleotidases; (e) ligates specific plasma membrane receptors, a feature that confers specificity to its signaling. These properties allow the generation of an extracellular messenger characterized by (a) very low background noise and thus high signal-to-noise ratio; (b) rapid diffusion through the aqueous tissue interstitium; (c) rapid signal shut-off to avoid overstimulation or receptor desensitization [11].

Over the years, a role for eATP has been identified in several, different physiological and pathological conditions such as [12]: glial-neuron interaction [13], sensory transmission [14], hormone secretion [15], disorders of central nervous system [16], cardiovascular diseases [17], infection [18], inflammation [19] and cancer [20].

## 2. Mechanisms of ATP Release

Intracellular ATP generated by glycolysis and oxidative phosphorylation can be released into the TME in a non-regulated fashion in response to various cell stress- and cell death-inducing conditions, such as hypoxia, cytotoxic agents, autophagy or plasma membrane damage [21,22,23,24]. In spite of the high eATP levels of the TME, the much higher intracellular ATP concentration (in the 5–10 millimolar range) generates an outward-directed gradient for ATP, thus facilitating passive efflux. In addition to the chemical gradient, the electric gradient (the plasma membrane potential, negative inside, facilitates efflux of a negatively charged molecule such as ATP) also supports passive ATP release. ATP released by stressed and necrotic cells acts as a danger signal generating a pro-inflammatory microenvironment that promotes the recruitment of immune cells to the damage sites [25,26]. It is likely that eATP is the prototypical and most widely diffused damage-associated molecular pattern (DAMP).

In addition to passive leakage, it is now generally agreed that most cells are able to actively release ATP [27]. Cytosolic vesicles, thanks to the vesicular nucleotide transporter (VNUT or SLC17A9), accumulate ATP, which is then rapidly released via stimulated or constitutive exocytosis [28]. The secretory vesicles VNUT transporter accumulates ATP at the expense of the proton electrochemical gradient (positive inside) generated by the vacuolar ATPase (V-ATPase) [29]. Vesicular exocytosis is regulated by intracellular Ca^2+^ levels and by the soluble N-ethylmaleimide-sensitive factor attachment protein receptor (SNARE) [30]. Release of ATP-laden vesicles is dependent on the PI3K/Rho/ROCK pathways and on the reorganization of the actin cytoskeleton and, of relevance in tumors, is promoted by hypoxia [31,32]. Release of ATP by stimulated exocytosis occurs in various cell types, including neurons and secretory cells [33,34]. Anecdotal evidence suggests that T lymphocytes may also release ATP by a similar mechanism; thus, vesicular release could be important in setting ATP levels in the TME during T lymphocytes activation [35]. Plasma membrane-derived microvesicles may be an additional mechanism of ATP release.

Besides vesicular release, numerous studies have shown that ATP can be released through non-exocytotic conductive pathways. ATP-binding cassette (ABC) transporters that hydrolyze ATP to support the transmembrane movement of different molecules have been proposed to function as ATP-releasing pathways [36]. Mammalian ABC transporters are integral membrane proteins comprised of 12 membrane-spanning domains and two conserved cytoplasmic domains, which bind and hydrolyze ATP [37]. Among them, multidrug resistance protein 1 (MDR1, also known as P-glycoprotein) is associated with ATP channel activity. The first demonstration that MDR1 may be implicated in ATP transport originated from the observation that ATP released from CHO (Chinese hamster ovary) cells and human lung tumor cells was proportional to the level of MDR1 expression. Moreover, the transfection of the wild-type *mdr1* gene increased the amount of released extracellular ATP compared to untransfected controls [38]. Cystic fibrosis transmembrane conductance regulator (CFTR) was also suggested to be an ATP-releasing pathway [39], but subsequent studies were unable to confirm this finding [40,41]. Over the years, other anion channels were recognized as ATP-permeable channels, including chloride ion channels [42] and volume- and voltage-dependent anion channels (VDAC) [43]. Currently, five families of channels are thought to mediate various forms of physiological and pathophysiological ATP release: connexin hemichannels, pannexin 1 (PANX1), calcium homeostasis modulator 1 (CALHM1), volume-regulated anion channels (VRACs, also known as volume-sensitive outwardly rectifying (VSOR) anion channels) and maxi-anion channels (MACs) [44] (Figure 1). Connexins and pannexins share similar structural features, with N- and C- terminal cytoplasmic domains, four membrane-spanning segments and both intracellular and extracellular loop domains [45].

Connexin and pannexin proteins assemble to form hexameric membrane structures called connexons and pannexons, respectively, that mediate the release into the extracellular space of small molecules, including ATP, glutamate and others with MW below 1–2 kDa [46,47]. The main difference between these two channel-forming proteins is that connexins can form gap junctions and hemichannels, while pannexins only form hemichannels [48]. Gap junctions allow direct communication between adjacent cells, while undocked hemichannels mediate the release of cytoplasmic components [49,50]. Connexins, of which more than 20 isoforms have been currently identified, are widely distributed [49]. In 1998, Nedergaard and co-workers provided the first evidence for the involvement of connexins in cellular ATP release. They stably transfected C6 glioma cells (lacking endogenous gap junctions) with connexin 43 (Cx43) or connexin 32 (Cx32), showing that the Cx43^+^ and Cx32^+^ C6 clones released more ATP compared to wild-type C6 cells [51]. Moreover, they also noted the potentiation of ATP release in other cells transfected with connexins, including Cx43-, Cx32-, Cx26- and Cx30-overexpressing HeLa cells and Cx32-overexpressing U373-MG human glioblastoma cells. Other connexin isoforms, such as connexin-26, connexin-37 and connexin-36, were shown to mediate ATP release [52], although ATP permeability was directly demonstrated only for connexin-43 hemichannels [46]. In monocytes and macrophages, connexin-43 activation and the associated ATP release may be regulated by hypoxia, changes in intracellular calcium concentration, reactive oxygen species (ROS), nitric oxide (NO) and stimulation of TLR2 and TLR4 [53,54,55].

The human pannexin family consists of three members: pannexin-1, -2 and -3 [52]. Pannexin-1 is expressed in different excitable and non-excitable cells, whereas pannexin-2 and pannexin-3 are restricted to the brain and skin/bone, respectively [56]. There is ample evidence to support the function of pannexin-1 as a plasma membrane channel and its function as an ATP release channel [57]. The most direct evidence that pannexin-1 is an ATP-permeable channel was obtained using *Xenopus* oocytes injected with human pannexin-1 cRNA [47]. The opening and activation of pannexin-1 channels are mediated by multiple events, such as intracellular Ca^2+^ increase, redox potential changes, mechanical stress and P2X7 receptor (P2X7R) activation [47,58,59]. In 2014, Dahl and co-workers proposed a model whereby pannexin-1 forms two open channel conformations depending on the mode of activation: a large conductance, ATP-permeable, conformation induced by many physiological stimuli (including extracellular K^+^, intracellular Ca^2+^, low oxygen tension) and an intermediate-conductance, ATP-impermeable, conformation activated by membrane depolarization [60]. Thus, they hypothesized two different channel states associated with the two biophysical channel proprieties, suggesting that the large conductance may be an essential requirement for the ATP permeability. However, a more recent study confuted this conclusion showing that pannexin-1 activated by truncation of the carboxyl-terminal auto-inhibitory region exhibits intermediate conductance currents associated with ATP release [61].

Channels belonging to the calcium homeostasis modulator (CALHM) family have been proposed to support ATP release. To date, only CALHM1 out of the six members (CALHM1-6) of the family has been identified as a functional ATP-permeable ion channel. CALHM1 acts as a pore-forming subunit of a plasma membrane, voltage-gated, non-selective ion channel with a pore large enough to accommodate ATP [62,63]. ATP release associated with CALHM1 expression has been demonstrated in vitro and in vivo [64]. Heterologous expression of CALHM1 in HeLa cells, COS cells and *Xenopus* oocytes led to ATP release into the extracellular space in response to different stimuli, including membrane depolarization and lowering of extracellular Ca^2+^ concentration [65].

In response to hypotonic cell swelling, cells activate several processes to restore normal cell volume, among which the major mechanism is the conductive efflux of organic and inorganic osmolytes through the anion-selective channels VRACs (also known as VSORs) [66].

VRACs are ubiquitously expressed in many tissues where they can be activated by hypotonic cell swelling via a reduction in the cytoplasmic ionic strength, as well as by the activation of plasma membrane receptors, including purinergic, epidermal growth factor and bradykinin receptors [67,68]. Although pharmacological evidence does not support the involvement of VRAC channels in swelling-induced ATP release [69,70], ATP permeability of VRACs was recently demonstrated by luciferin/luciferase-based measurements of ATP release from *Xenopus* oocytes exposed to hypotonic stress [71].

In addition to VRAC channels, maxi-anion channels (MACs) are also directly activated by cell swelling and involved in the ATP release mechanism [72]. MACs are voltage-dependent, large-conductance, ATP-permeable anion channels present in every cell type and activated by different stimuli, including hypoxia, high glucose and osmotic swelling [73,74]. The first evidence of MACs permeability to ATP stemmed from observations in mouse mammary C127i cells suggesting that ATP interacts with a site inside the pore lumen [75]. This report was later supported by the estimation of the pore entrance size, confirming that the pore entrance is wide enough to accommodate ATP molecules [76]. Moreover, Okada and co-workers discovered that SLCO2A1 is the core pore-forming component of MACs and showed that ATP release is potentiated by the heterologous expression of SLCO2A1 in HEK293T cells, suggesting that MAC channels made by SLCO2A1 are permeable to ATP [77].

An additional pathway for ATP release is the P2X7R. Besides a cation-selective channel, the P2X7R can generate a large, non-selective pore (macropore) permeable to ATP and other molecules up to 900 Da. Although it was originally thought that P2X7R-associated pore opening occurs after the opening of the P2X7R ion channel, and thus usually only after prolonged stimulation, it is now clear that pore opening occurs with no delay after P2X7R gating, thus allowing early fluxes of low as well as high MW molecules [78]. Over the years, solid evidence has accumulated to foster the role of the P2X7R in ATP release in response to a variety of stimuli, a finding further supported by the observation that P2X7R gating causes intracellular ATP depletion [79,80,81,82].

Besides ion channels, the eATP level in the extracellular space can also increase thanks to additional mechanisms. Adenylate kinase (AK) catalyzes the reversible reaction ATP + AMP ↔ 2ADP. Different AK isoforms are present in the cytosol (AK1), in the mitochondrial intermembrane space (AK2), in the mitochondrial matrix (AK3) and in the nucleus (AK6). Recently, an ecto-AK activity on different cell types, including human hepatocytes and leukemic cell lines, has been identified, suggesting a role in the regulation of eATP levels [83,84,85]. In addition, nucleoside diphosphate kinases (NDPK/NME/NM23) are also responsible for the conversion of extracellular AMP and ADP to ATP. These enzymes act as cell-surface ectoenzymes expressed on the plasma membrane fueling trans-membrane nucleotide transfer. Plasma membrane NDPK responsible for extracellular ATP synthesis has been identified in glioma cells, lymphocytes and hepatocytes [83,84,86,87].

Finally, the F1F0 ATP synthase has been proposed to participate in extracellular ATP synthesis. The F1F0 ATP synthase is the mitochondrial enzyme that couples ATP synthesis to the transmembrane electrochemical proton gradient. Although the contribution of F1F0 ATP synthase to extracellular ATP synthesis is not fully understood, plasma membrane localization of ATP synthase has been identified in adipocytes, human keratinocytes and tumor cell lines [88,89]. Thus, non-lytic pathways for ATP release are expressed by both immune and tumor cells, suggesting that regulated release is the main pathway for eATP accumulation in the TME.

## 3. Detection of Extracellular ATP in the TME

It is an established notion that ATP is one of the main components of the tumor microenvironment (TME), where it affects cancer cell proliferation, motility and dissemination and antitumor immune response [10,90]. Several techniques to measure eATP in vitro and in vivo have been developed over the years [91]. Dubyak and co-workers proposed a method to measure ATP in the pericellular space by using a cell-surface-bound luciferase. Firefly luciferase was fused in frame with the immunoglobulin G (IgG)-binding domain of protein A from *Staphylococcus aureus*, thus allowing binding of the chimeric protein to IgG absorbed onto the cells surface [92]. More sophisticated approaches exploited atomic force microscopy to measure local ATP concentration [93], fluorescence microscopy for real-time ATP measurement by the two-enzyme system [94] or HPLC-based methods [95]. Other in vitro methods to measure ATP include the patch-clamp technique [96] and enzyme-coated platinum microelectrodes. Microelectrodes can also be used to monitor eATP changes in vivo [97]. More recently, additional techniques have been proposed, such as an assay that measures the conversion of NADP^+^ into NADPH in the presence of ATP by fluorescence microscopy [98], a sensor based on malonyl-coenzyme A synthetase that undergoes a conformational change upon ATP-binding, thus causing an increase in fluorescence intensity [99] and ratiometric, Förster resonance energy transfer (FRET)-based fluorescent indicators [100,101]. Most of these techniques have high sensitivity, but are in general of difficult if not impossible application in vivo. The probe that provides most reliable measurements of the eATP concentration in vivo, and to a certain extent in vitro, is a plasma membrane-expressed luciferase, named pmeLUC (plasma membrane luciferase), developed in our laboratory [79]. Use of the pmeLUC probe has unequivocally demonstrated that eATP TME levels are in the range of high micromolar level (50–200 µmol/L), whereas in healthy tissues eATP concentration is submicromolar (likely about 10–100 nmol/L) [102,103]. PmeLUC is a chimeric protein engineered by appending to the *Photinus pyralis* luciferase cDNA a N-terminal leading sequence and a C-terminal glycosyl phosphatidylinositol (GPI) anchor from the human folate receptor [79]. Thanks to this modification, the pmeLUC probe is targeted to the external side of the plasma membrane, thus allowing ATP measurement in the extracellular space. This probe can be expressed in all cell types amenable to transfection, which can be either used to generate a primary tumor or can be inoculated into a tumor (or into the peritumor space) to monitor ATP concentration [104,105].

Over the years, the pmeLUC probe has been used to measure the eATP concentration in the TME of many different experimental tumors (for example, neuroblastoma, human ovarian carcinoma, human melanoma, mouse colon carcinoma and liver metastasis of human colon carcinoma in mice), consistently showing that the TME is rich in extracellular ATP [103,106,107,108].

## 4. Role of Extracellular ATP in the TME

In the TME, extracellular ATP is degraded by different ecto-nucleotidases, chiefly CD39 and CD73. The ecto-nucleoside triphosphate diphosphohydrolase CD39 hydrolyzes ATP to ADP and AMP, while ecto-5′-nucleotidase CD73 hydrolyzes AMP to adenosine, which is further degraded to inosine by adenosine deaminase (ADA). CD39 and CD73 affect tumor growth thanks to their ability to produce adenosine, which promotes immunosuppression in the TME via adenosine P1 receptors [109]. CD39 is highly expressed by dendritic cells (DCs), tumor-infiltrating Th17 lymphocytes and M2 macrophages [110,111]. CD73 is overexpressed by many tumors and expressed by a subpopulation of human T and B lymphocytes, stromal cells and dendritic cells [112,113]. CD39 and CD73, which are both upregulated in the hypoxic TME, impair the antitumor immune response and facilitate tumor progression [109]. Several studies show that *cd39*^−/−^ or *cd73*^−/−^ mice develop spontaneous inflammatory bowel or lung injury and that *cd73*^−/−^ mice are resistant to experimental metastasis, suggesting that the CD39/CD73 axis is important for inflammation and therefore for the ATP pro-inflammatory activity that promotes antitumor immunity in the TME [114,115]. Probably, soluble or microvesicle-associated ATPases accumulate in the TME, where they cooperate with plasma membrane ecto-nucleotidases in ATP degradation and adenosine accumulation. Adenosine, actively generated by CD73, ligates the P1 purinergic receptors (P1Rs), divided into four subtypes: A1R, A2AR, A2BR and A3R. P1Rs are G protein-coupled receptors, and their activation triggers stimulation or inhibition of adenylyl cyclase (AC) (depending on the receptor subtype), modifying intracellular cAMP content. Moreover, A1R, A2BR and A3R stimulation also triggers phospholipase C (PLC) activation and thus release of Ca^2+^ from intracellular stores. Due to the differential signaling induced and the diverse expression of P1 receptors on cancer cells, adenosine may trigger both tumor growth stimulation and inhibition [116]. A1Rs display protumoral effects increasing melanoma chemotaxis and breast cancer proliferation, but on the other hand, antiproliferative effects have been observed in glioblastoma, colon cancer and leukemia cells. A2ARs may enhance proliferation in breast cancer or trigger cell death in melanoma, while A2BRs and A3Rs are involved in metastatic spreading and apoptosis [116,117,118,119]. However, adenosine is well known for its immunosuppressive activity in the TME. Of the four P1R subtypes, A2AR is the main adenosine receptor expressed by immune cells, including dendritic cells (DCs), macrophages, B and T cells. A2AR stimulation inhibits antigen presentation by DCs, drives M2 macrophage differentiation and strongly influences T cell activation. This receptor mainly counteracts TCR-mediated signaling of T lymphocytes by increasing intracellular cAMP levels and promoting the activity of protein kinase A (PKA), which attenuates T cell proliferation and inhibits inflammatory cytokine production and effector functions [120,121].

Extracellular ATP acts on tumors via specific plasma membrane receptors. Virtually all cancer and immune cells express P2 receptors and are sensitive to extracellular ATP. P2 receptors are divided into two subfamilies: P2Y (P2YR) and P2X (P2XR) receptors. P2YRs are G-protein coupled metabotropic receptors that comprise eight members (P2Y1R, P2Y2R, P2Y4R, P2Y6R, P2Y11R, P2Y12R, P2Y13R and P2Y14R), while P2XRs are ionotropic receptors that include seven members (P2X1-7R). P2X subunits assemble to form homo- or hetero-trimeric cation-selective channels [122]. Extracellular ATP acting at P2Rs may sustain cell proliferation or trigger cell death via different pathways, depending on the ATP concentration and on the level of ectonucleotidases and specific P2Rs expressed by immune and tumor cells [123]. Among P2YRs, P2Y1R, P2Y2R and P2Y6R support cell growth of healthy and cancer cells by activating the PI3K-AKT and ERK-MAPK pathways or by exacerbating DNA damage in tumor cells [124], thus eATP in the TME may promote tumor proliferation and support invasiveness and metastatic spreading via P2YRs. In addition, in several cancer cell types (human melanoma MDA-MB-435s cells, human PC-3M prostate cancer cells, PC8 lung carcinoma and human T47D breast cancer cells) extracellular ATP triggers metalloproteinase (MMP) and cathepsin secretion via P2X7R stimulation, facilitating local invasion and metastatic spreading (Figure 2). Epithelial-mesenchymal transition (EMT) is also promoted in breast and prostate cancer cells via P2Y2R activation [125,126,127].

Over the years, several studies have shown that the P2X7R has growth-promoting activity and supports tumor progression [128,129,130], although this receptor is widely known for its cytotoxic activity. This highlights an interesting paradox, as the high eATP concentration in the TME should be sufficient to trigger P2X7R-mediated cytotoxicity, but surprisingly, cancer cells seem to be strikingly refractory. The reason for this refractoriness is unknown, but it has been suggested that it may at least in part be due to the uncoupling of P2X7R activation from the intracellular death machinery [129], or otherwise, the high plasma membrane cholesterol may inhibit P2X7R pore opening, while leaving the growth-promoting activity of P2X7 channel unperturbed [131]. Besides the P2X7R, P2X3R, and P2X5R overexpression has also been implicated in tumor cell proliferation and survival [132,133].

Previous data showed that eATP acting mainly at P2X7R, P2X1R and P2YRs is a stimulus for intracellular energy metabolism [134,135,136]. Basal activation of P2X7R (and possibly other P2Rs) causes an elevation in cytoplasmic and mitochondrial Ca^2+^ concentration, stimulating oxidative phosphorylation and promoting ATP production [136]. P2X7R stimulation also upregulates expression of the glucose transporter GLUT1 and other glycolytic enzymes involved in aerobic glycolysis, supporting the generation of an acid, immunosuppressive TME [137] (Figure 3).

It is now clear that ATP concentration in the TME shapes the cellular and biochemical composition of the TME in different ways. The basic assumption is that low ATP levels promote tumor proliferation and immunosuppression, while high ATP levels activate infiltrating inflammatory cells and promote antitumor immunity [9]. However, eATP acting at P2YRs (and possibly at P2X7R as well) is also known to drive recruitment of inflammatory cells in the TME, and therefore is in principle responsible for the highly immunosuppressive heavy inflammatory infiltrate, tumor-associated macrophages (TAM), T_reg_ cells and myeloid-derived suppressor cells (MDSCs) included [138,139]. In addition, ATP may fuel a direct antitumor immune response by depleting T_reg_ cells via P2X7R-mediated cytotoxicity or via stimulation of DCs, thus triggering IL-1β release and potentiating tumor antigen presentation to CD4^+^ and CD8^+^ lymphocytes [140,141,142,143] (Figure 2).

## 5. Extracellular ATP as a Target for Cancer Therapy

Due to its key role in tumor-host interaction, the most obvious target for cancer therapy would be eATP itself. Attempts to reduce the TME eATP concentration might be beneficial as this might decrease adenosine generation and growth stimulation of cancer cells, or alternatively, it might be rather advisable to further increase its concentration to exploit ATP-dependent cytotoxicity. In the mid ‘70 s, Rozengurt and Heppel and Landry and Lehninger showed that eATP causes permeability changes of the plasma membrane and severely upset intracellular ion balance [144,145]. Later, our own laboratory carried out an extensive investigation of the mechanisms responsible for eATP-dependent cell death [146,147,148,149]. Reports on the anticancer activity of eATP have accumulated over the years. Rapaport first showed that the addition of eATP inhibited cancer cell growth by arresting the cell cycle in the S phase [150]. Moreover, he demonstrated that intraperitoneal injection of ATP (50 mM) into tumor-bearing mice effectively reduced tumor size [151]. Further in vivo studies performed in human prostate cancer xenograft showed that daily intraperitoneal administration of eATP (25 mM) causes a significant tumor regression [152]. In addition, clinical studies reported that intravenous administration of ATP was well tolerated by cancer patients, leading to an improvement of tumor-associated cachexia and overall health conditions, suggesting that eATP in combination with other treatments may not only help to reduce tumor size but may also reduce systemic side effects [153,154].

Among P2Rs, the P2X7R is an appealing target. However, a substantial difficulty in devising P2X7R-targeting strategies for cancer therapy is the very steep eATP activation curve of the P2X7R that precludes the controlled activation of this deadly receptor using eATP or pharmacological ATP analogs. Therefore, the use of positive allosteric modulators, compounds that potentiate P2X7R activation in the presence of relatively low eATP concentrations, such as polymyxin B, ginsenosides and clemastine [155,156,157], may provide novel strategies to enhance P2X7R-mediated cytotoxicity in cancer. Hyperthermia has also been shown to induce cancer cell death via P2X7R activation, thus suggesting that combining eATP with hyperthermia could be a clinically viable option [158]. An interesting extension of P2X7R-targeting in cancer therapy comes from the finding that eATP increases plasma membrane permeability for cytotoxic molecules (e.g., doxorubicin, ethidium bromide, thiocyanate) via P2X7R pore opening [159,160]. More recently, a mechanism has been proposed by which shock wave treatment of human osteosarcoma U2OS cells induces the efflux of intracellular ATP, which promotes the intake of methotrexate (MTX) by altering plasma membrane permeability via P2X7R activation, thus leading to MTX-induced apoptosis [161].

Despite the beneficial effect of eATP for the treatment of cancer, its major disadvantage is the rapid degradation by ecto-nucleotidases, thus requiring high doses to reach a therapeutic effect, as shown by the observation that continuous intravenous infusion of ATP does not increase the eATP content [162]. A sophisticated strategy to prevent fast eATP degradation has been proposed by Diaz-Saldivar and Huidobro-Toro. These authors synthesized highly biocompatible and biodegradable albumin nanoparticles (ANPs) loaded with ATP and coated with erythrocyte membranes (EMs). This treatment controls the rate of ATP release and extends the nanoparticle circulation time. Incubation of these ATP-laden nanoparticles with HeLa and Hek293 cell cultures produced a constant and controlled release of ATP, thus enhancing the antiproliferative activity of this nucleotide [163]. A similar strategy was developed using a pH-sensitive nanoplatform made up of chitosan (Cs) and mesoporous hydroxyapatite (HAP), which was used for ATP delivery to tumor cells. This nanoplatform induced a high rate of apoptosis and slowed down tumor cell growth [164]. Another option to increase eATP in a controlled fashion may be CD39 and/or CD73 targeting [165,166] (Table 1). Recent studies demonstrated that the use of anti-CD39 antibodies boosts the eATP-P2X7-inflammasome-IL-18 axis, thus resulting in a decrease of macrophage recruitment and an increase in effector CD8^+^ T cells in the TME [167]. On the same line, CD39 and CD73 suppression using antisense oligonucleotides (ASOs) has been shown to prevent eATP degradation and to rescue T cell proliferation, thus promoting an efficient antitumor immune response [168].

In the TME, extracellular ATP has a central role in the mechanism of immunogenic cell death (ICD), which is an atypical type of cell death associated with DAMP (danger-associated molecular pattern) release (e.g., ATP or high-mobility group protein B1, HMGB1), or DAMP exposure on the plasma membrane (e.g., calreticulin, type I interferons) [169]. A key step of ICD is eATP-dependent P2X7R stimulation and the ensuing IL-1β release via NLRP3 inflammasome activation. IL-1β stimulates dendritic cells (DCs) activation and tumor antigen presentation to CD4^+^ and CD8^+^ T lymphocytes, thus promoting an efficient antitumor immune response [140]. Several chemotherapeutics, such as mitoxantrone, anthracyclines and oxaliplatin, trigger ICD and the associated ATP release via canonical pannexin 1 channels or P2X7R. γ-irradiation has been reported to trigger P2X7R-dependent ATP release from B16F10 mouse melanoma cells, suggesting that this receptor plays a role in radiation-induced ICD [82]. Recently, preclinical and clinical studies have shown the efficacy of the combination of radiotherapy-induced ICD and immune checkpoint inhibitors for cancer treatment [170]. An overview of clinical trials targeting eATP metabolism (i.e., CD39 or CD73 ecto-nucleotidases), or exploiting eATP to confer tumor selectivity to therapeutic antibodies, is shown in Table 1.

Autophagy, a key component of the adaptive cell survival response, likely participates in setting extracellular ATP levels in the TME. Autophagy-dependent ATP release is required for the antitumor effects promoted by chemotherapy-induced cell-death [107] and by the caloric restriction mimetic hydroxycitrate [108]. A novel approach to exploit extracellular ATP for antitumor therapy has been proposed by Igawa and co-workers (Figure 4). They generated an anti-CD137 switch antibody (STA551) that exerts agonistic activity only in the presence of high ATP concentration (above 100 μmol/L), a level close to that of TME eATP. The STA551 antibody demonstrated potent and selective activity in different mouse and human tumor models without generalized immune system activation. Moreover, this antibody showed a synergic antitumor effect with anti-PD-L1 and increased CD8^+^ T cell proliferation and infiltration (Table 1). This suggests the possible use of STA551 in combination with other treatments, such as chemotherapy or administration of starvation mimetic compounds, to boots antitumor immune response [171].

On the other hand, high ATP concentration could lead to generation of large amounts of adenosine and therefore promote immunosuppression. An alternative strategy to decrease eATP levels in the TME, avoiding excess adenosine generation, is to inhibit ATP release. Suitable targets are the pannexin 1 channels or the P2X7R itself. Pannexin 1 inhibition with different non-selective compounds, such as carbenoxolone, mefloquine or probenecid, decreases ATP release, and its blockade has been shown to reduce cancer cell survival and to inhibit metastatic spreading [172]. Several small molecules P2X7R blockers, including AZD9056, AZ10606120 and GSK1482160, were reported to exhibit antitumor activity in multiple experimental tumor models [173,174]. Genetic disruption or pharmacological inhibition of P2X7R not only reduces ATP release but also strongly impairs energy metabolism, reducing cancer cell proliferation and invasiveness [134].

Targeting ATP production using antiglycolytic agents seems to be an effective therapeutic approach to suppress cancer progression [175]. A recent study demonstrated that the cardiac glycoside ouabain affects glycolysis and greatly decreases mitochondrial oxidative phosphorylation (OXPHOS) in human lung cancer A549 cells and human breast cancer MCF7 cells, thus depleting ATP production and contributing to cancer cell cytotoxicity via AMPK activation [176].

The opportunistic uptake of extracellular nutrients via multiple endocytic mechanisms has been described as a primary hallmark of cancer metabolism [177]. Chen and co-workers showed that eATP is internalized by various cancer cells by micropinocytosis to elevate intracellular ATP concentration, promote cell growth and survival and enhance resistance to different anticancer drugs [178,179]. Furthermore, elevated intracellular ATP upregulates signal transduction, including signals involved in epithelial-mesenchymal transition (EMT), migration and invasion. Knockout of the sorting nexin 5 (*SNX5)* gene (a protein very important for macropinocytosis) lowers intracellular ATP levels and inhibits cancer cell proliferation and cell migration, confirming the important role of macropinocytosis in eATP-mediated tumorigenesis and metastasis [180]. These findings could also shed light on the mechanism by which cancer cells grow faster than normal cells despite a largely inefficient ATP synthesis. The presence of high levels of eATP in the TME might be a source of energy for cancer cells via eATP internalization, that in turn leads to intracellular ATP increase, or via the generation of high energy molecules using eATP as a phosphate donor [181]. Finally, a recent study demonstrated that treatment with apyrase (an ATP hydrolase) in combination with SOX-9 (sex-determining region Y-box 9) knockdown decreased tumor growth and enhanced drug sensitivity in mice [182], confirming that reducing eATP levels and inhibiting eATP-dependent responses are a viable anticancer approach.

## 6. Conclusions

For many years, antitumor therapy was based on the administration of highly toxic treatments aimed at killing each and every cancer cell, with the side effect of substantial damage to the host tissues. Today, the increasing awareness of the complex network of information exchange occurring in the TME provides a unique opportunity to fight tumors by taking advantage of the intrinsic defenses of the host. Several in vitro and in vivo studies show that extracellular nucleotides, mainly ATP, affect tumor growth and tumor-host interaction in multiple ways. Extracellular ATP can promote cancer cell growth and metastatic spreading or trigger cancer cell death, is involved in immune cell recruitment and activation, and can also promote immunosuppression by being degraded to adenosine. Over the past few years, several different pathways for ATP release into the extracellular milieu have been identified, and multiple plasma membrane receptors responsible for the ATP-dependent effects have been characterized. Several potent and highly selective pharmacological blockers have also been synthesized, among which P2X7R-targeting small molecule drugs and antibodies may be promising cancer therapies.

An appealing therapeutic approach may be to directly target eATP in the TME in the attempt to reduce its levels and thus inhibit adenosine generation or cancer cell growth, or further increase its concentration to exploit ATP-dependent cytotoxicity. Different strategies have been proposed to regulate ATP release and accumulation in the TME, thus inducing eATP-dependent cancer cell-death or promoting antitumor immune response. At the same time, multiple approaches have been developed to decrease eATP content and inhibit ATP production, thus enhancing anticancer drug sensitivity and reducing immunosuppression. However, a word of caution is needed here. Low eATP levels have a growth-promoting activity; therefore, manipulations aimed at lowering TME eATP may also unwillingly end up with the undesired effect of promoting tumor growth.

In conclusion, eATP is a major biochemical constituent of the TME with a fundamental role in tumor-host interaction and an appealing target for innovative anticancer therapy.

## Figures and Tables

**Figure 1 cells-09-02496-f001:**
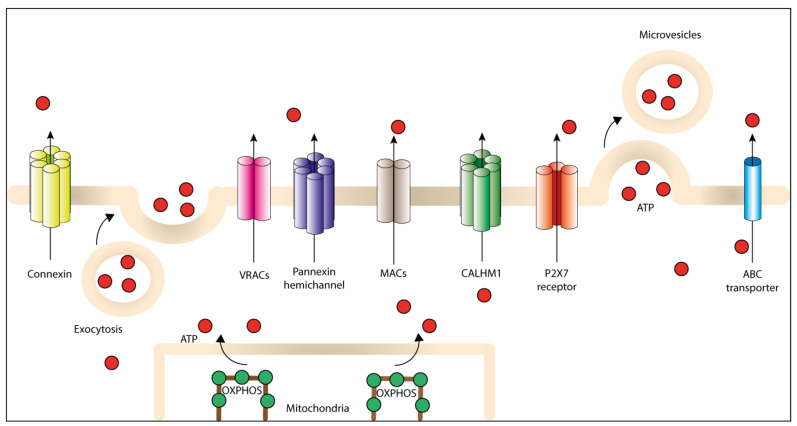
Different pathways for regulated ATP release into the tumor microenvironment (TME). ATP generated inside the cell can be actively released through plasma membrane-derived microvesicles, vesicular exocytosis or different non-exocytotic conductive pathways, including specific ATP-binding cassette (ABC) transporters, the P2X7R, connexin and pannexin channels, calcium homeostasis modulator 1 (CALHM1) channel, volume-regulated ion channels (VRACs) and maxi-anion channels (MACs).

**Figure 2 cells-09-02496-f002:**
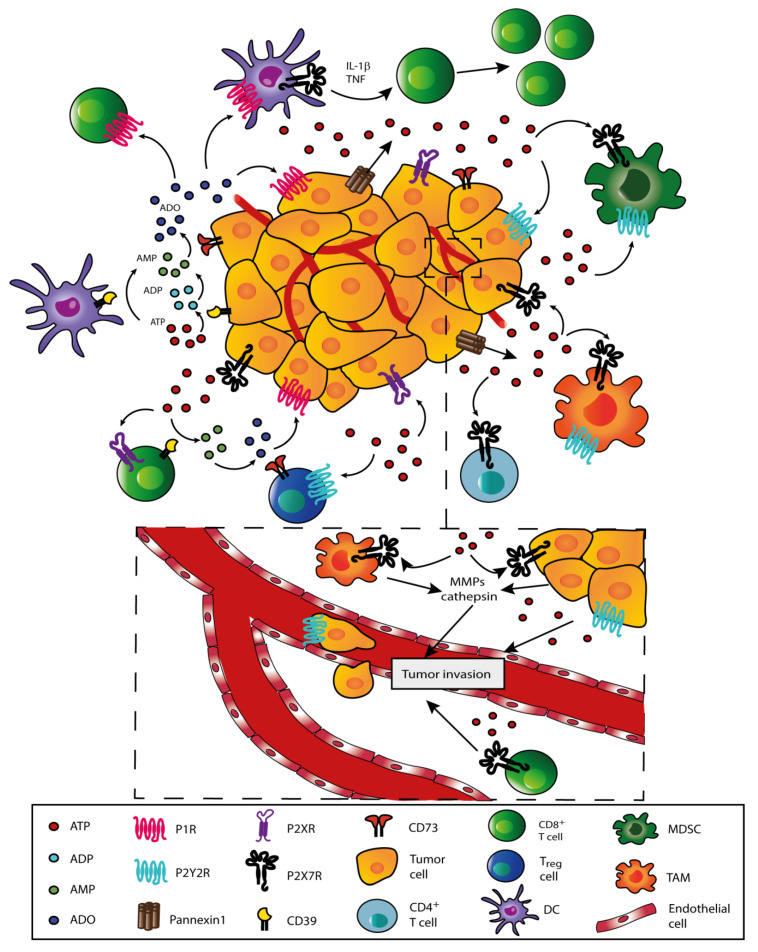
Extracellular ATP shapes the TME. ATP is released into the tumor microenvironment (TME) in a non-regulated or regulated fashion. Extracellular ATP can promote immunosuppression or support antitumor immunity depending on its concentration and specific receptors expressed by immune and cancer cells. ATP is degraded by ecto-nucleotidases (CD39 and CD73) to generate ADP, AMP and adenosine (ADO), which promote immunosuppression via adenosine receptors P1 (P1Rs) (mainly A2AR and A2BR). A2AR stimulation inhibits antigen presentation by dendritic cells (DCs) and impairs cytotoxic T lymphocytes functions. Extracellular ATP acting at P2Y receptors (P2Y1R, P2Y2R or P2Y6R) and P2X receptors (mainly P2X7R) supports tumor cell survival and proliferation, but at the same time drives recruitment and activation of immune cells such as CD8^+^ and CD4^+^ T lymphocytes, T_regs_, tumor-associated macrophages (TAMs) and myeloid-derived suppressor cells (MDSCs). In addition, ATP activates DCs to promote the release of pro-inflammatory cytokines, such as IL-1β and tumor necrosis factor (TNF), and potentiate tumor antigen presentation. ATP-activated DCs increase CD8^+^ T cell responses, thus supporting antitumor immunity.In the TME, eATP triggers via P2X7R the release of metalloproteinases (MMPs) and cathepsin from tumor cells, T lymphocytes and macrophages. MMPs degrade the extracellular matrix and support tumor cell invasion and metastatic spreading. ATP-mediated activation of P2YR, e.g., P2Y2R, drives the formation of pseudopodia and facilitates tumor cell migration across vessel endothelium.

**Figure 3 cells-09-02496-f003:**
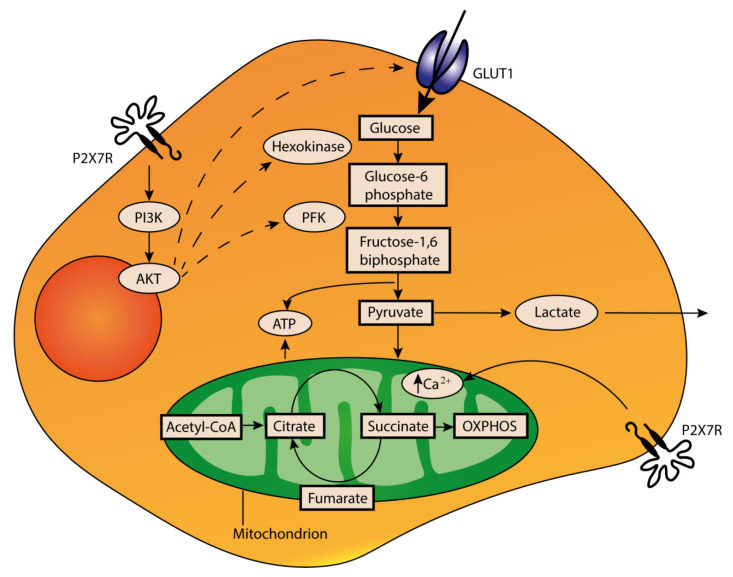
Stimulation of cancer cell metabolism. Extracellular ATP acting at P2X7R causes an increase in the mitochondrial Ca^2+^ level, stimulates oxidative phosphorylation (OXPHOS) and promotes ATP generation. At the same time, P2X7R activation upregulates via PI3K-AKT expression of the plasma membrane glucose transporter GLUT1 and of several enzymes of the glycolytic cascade. This generates ATP and, at the same time, increases the production of lactate, which causes acidification of TME.

**Figure 4 cells-09-02496-f004:**
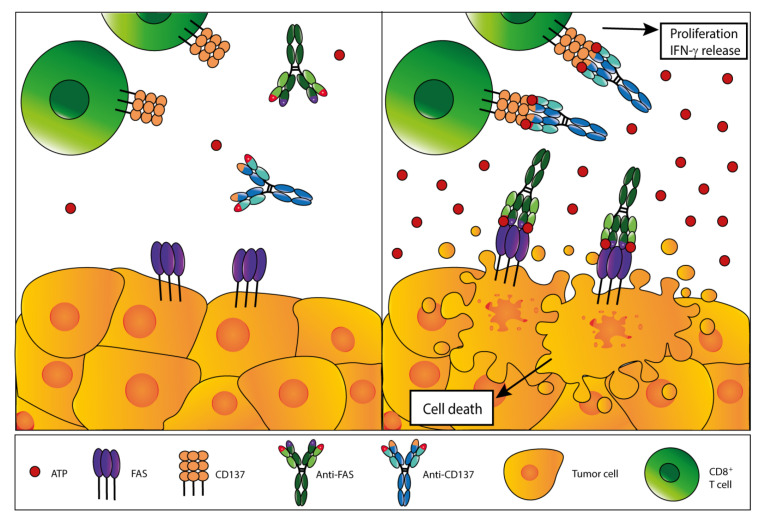
Strategies to exploit the elevated eATP concentration in the TME for antitumor therapy. A novel anti-CD137 monoclonal antibody was generated by Igawa and co-workers [171] that binds the antigen only in the presence of ATP levels close to those found in the TME (**left panel**). Under these conditions, and only under these conditions, this antibody binds and activates the costimulatory receptor CD137 expressed by CD8^+^ T cells, thus promoting T cell proliferation and cytokines release (e.g., IFN-γ), which boosts antitumor immune response (**right panel**). In the same way, we can hypothesize the engineering of an antibody that binds and activates suicide receptors on tumor cells, such as the FAS receptor, that is selectively activated by high ATP in the TME (**right panel**).

**Table 1 cells-09-02496-t001:** Overview of clinical trials targeting eATP in the TME.

Study	Identifier Code	Target
Study of SRF617 in patients with advanced solid tumors	NCT04336098	CD39
TTX-030 in combination with immunotherapy and/or chemotherapy in subjects with advanced cancers	NCT04306900 NCT03884556	CD39
A study of the CD73 inhibitor LY3475070 alone or in combination with Pembrolizumab in participants with advanced cancer	NCT04148937	CD73
CPI-006 alone and in combination with ciforadenant and pembrolizumab for patients with advanced cancer	NCT03454451	CD73
Study of TJ004309 in combination with atezolizumab in patients with advanced or metastatic cancer	NCT03835949	CD73
A phase I/Ib study of NZV930 alone and in combination with PDR001 and/or NIR178 in patients with advanced malignancies	NCT03549000	CD73
A phase I, open-label study to assess the safety, tolerability, pharmacokinetics and antitumor activity of MED19447 (oleclumab) in Japanese patients with advanced solid malignancies	JapicCTI-184194	CD73
An investigational immune-therapy study of experimental medication BMS986179 given alone and in combination with nivolumab	NCT02754141	CD73
A phase Ia/Ib study of STA551 as a single agent and in combination with atezolizumab in patients with solid tumors	JapicCTI-205153	CD137

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
