# Peer review of "Extracellular ATP: A Feasible Target for Cancer Therapy"

_cells, 2020, doi:10.3390/cells9112496_

Round 1

Reviewer 1 Report

The authors provide a well-written and comprehensive review of the mechanisms leading to elevated nucleotide concentrations in the tumor microenvironment (TME), and their effects on tumor growth and the anti-tumor immune response. The novelty of the present review lies in its focus on manipulating the extracellular ATP concentration in the TME in contrast to targeting nucleotide receptors or nucleotide-metabolizing ectoenzymes. The manuscript contains 4 informative Figures that visually transmit central concepts of the manuscript.

Nonetheless, some points should be addressed before the manuscript is finally accepted:

minor points:

  • in Section 3 (Mechanisms of ATP release) all of the depicted candidate mechanisms depicted in Figure 1 are discussed in detail with the exception of the ABC cassette transporter MDR1/Pgp. A few sentences detailing the structure of this protein and the experimental link to ATP release would be helpful
  • In spite of the generally high quality of English, the manuscript still contains a multitude of typological mistakes that should be corrected

major points:

  • the role of adenosine signaling and its receptors is not properly discussed. Although it is mentioned repeatedly that adenosine is associated with immune suppression, the mechanisms for this association are not discussed. In Figure 2, the P1Rs are depicted only on tumor cells, not on lymphocytes. However in the text there is no mention of the role of adenosine receptors on tumor cells, but it is repeatedly stated that P1Rs are “immunosuppressive”. Fig. 2 should be corrected to show P1Rs at least on T cells (if not on other immune cells as well). In the text, the different P1Rs (and the heterogeneity of their function) should be named, emphasizing the role of A2AR on T lymphocytes as a prime mechanism for immune suppression. Similarly, the role of P1Rs on tumor cells (including information on which of them might be expressed on tumor cells) should be discussed (or, alternatively, the P1Rs on tumor cells should be removed from Fig. 2).
  • Although the concept of manipulating eATP concentrations to inhibit tumor growth is innovative and novel, it is still a long way from clinical application. The authors discuss therapeutical strategies that both aim to increase or to decrease eATP levels. However, in Section 4 the authors correctly point out that: “The basic assumption is that low ATP levels promote tumor proliferation and immunosuppression, while high ATP levels activate infiltrating inflammatory cells and promote antitumor immunity”. 
A statement that manipulating eATP may move eATP levels into a range that is non-desirable, or an equivalent note of caution, should be added to the Conclusions.

Author Response

We thank the reviewer for his/her kind comments and his/her suggestions that we gratefully accept.

Minor points.

  • in Section 3 (Mechanisms of ATP release) all of the depicted candidate mechanisms depicted in Figure 1 are discussed in detail with the exception of the ABC cassette transporter MDR1/Pgp. A few sentences detailing the structure of this protein and the experimental link to ATP release would be helpful

        The ABC transporter is now briefly discussed at lines 83-90.

  • In spite of the generally high quality of English, the manuscript still contains a multitude of typological mistakes that should be corrected

        We have thoroughly revised the MS and hope that now English language is improved.

        Major points

  • the role of adenosine signaling and its receptors is not properly discussed. Although it is mentioned repeatedly that adenosine is associated with immune suppression, the mechanisms for this association are not discussed. In Figure 2, the P1Rs are depicted only on tumor cells, not on lymphocytes. However in the text there is no mention of the role of adenosine receptors on tumor cells, but it is repeatedly stated that P1Rs are “immunosuppressive”. Fig. 2 should be corrected to show P1Rs at least on T cells (if not on other immune cells as well). In the text, the different P1Rs (and the heterogeneity of their function) should be named, emphasizing the role of A2AR on T lymphocytes as a prime mechanism for immune suppression. Similarly, the role of P1Rs on tumor cells (including information on which of them might be expressed on tumor cells) should be discussed (or, alternatively, the P1Rs on tumor cells should be removed from Fig. 2).

        Adenosine receptor function is now dealt with at lines 249-267, with emphasis placed on the A2A            receptor. Fig. 2 has been modified  to show P1 receptor expression on T lymphocytes and tumor            cells.

  • Although the concept of manipulating eATP concentrations to inhibit tumor growth is innovative and novel, it is still a long way from clinical application. The authors discuss therapeutical strategies that both aim to increase or to decrease eATP levels. However, in Section 4 the authors correctly point out that: “The basic assumption is that low ATP levels promote tumor proliferation and immunosuppression, while high ATP levels activate infiltrating inflammatory cells and promote antitumor immunity”. 
A statement that manipulating eATP may move eATP levels into a range that is non-desirable, or an equivalent note of caution, should be added to the Conclusions.

        A note of caution has been added in Conclusions (lines 481-484).

Reviewer 2 Report

This is a timely and comprehensive review on the role of ATP in cancer and immune cell contribution to tumor microenvironment regulation. It provides an overview of the mechanisms of ATP release, detection, and enzymatic conversion. The review is well-written and summarizes the most important advances in the field. 

There are some suggestions that may be considered by the authors:

1. Part 2 - “Extracellular ATP is a major constituent of the TME” has the description of the extracellular ATP detection methods. Logically, this part may be placed after Part 3 – “Mechanisms of ATP release”. Considering the methodological content of Part 2, the title should also be modified.

2. Several additional references could be included in Part 3 “Mechanisms of ATP release” that describes the role of the cytoskeleton, PI3K/Rho/ROCK signaling axis, and regulated vesicular exocytosis. This mechanism might be relevant to tumor hypoxic microenvironment: 

Bodin and Burstock J. Cardiovasc Pharmacol, 38: 900-908, 2001 

Hirakawa et al, J Physiol, 558: 479-488, 2004

Koyama et al, J Physiol, 532: 759-769, 2001

Woodward et al, Am J Physiol, 297: L954-L964, 2009

In addition, the reference to purinergic regulation of intracellular metabolism should be included in Part 4 as it supports the idea of purinergic regulation of cancer metabolism.  

Lapel et al, Am J Physiol, 312:C56-C70, 2017.

4. Figure 2 size should be scaled down to fit the figure legend to the same page. 

3. A summary of clinical trial and/or ATP-based anti-cancer drugs would be helpful to appreciate the translational aspect of the review. 

Author Response

We wish to thank the reviewer for his/her kind comments and the usefiul suggestions.

  • 1. Part 2 - “Extracellular ATP is a major constituent of the TME” has the description of the extracellular ATP detection methods. Logically, this part may be placed after Part 3 – “Mechanisms of ATP release”. Considering the methodological content of Part 2, the title should also be modified.

        Part 2 "Extracellular ATP is a major constituent of the TME" has ben moved after Part 3 –                          “Mechanisms of ATP release”, and title has been modified to "Detection of extracellular ATP in the                TME"

  • 2. Several additional references could be included in Part 3 “Mechanisms of ATP release” that describes the role of the cytoskeleton, PI3K/Rho/ROCK signaling axis, and regulated vesicular exocytosis. This mechanism might be relevant to tumor hypoxic microenvironment: 

    Bodin and Burstock J. Cardiovasc Pharmacol, 38: 900-908, 2001 

    Hirakawa et al, J Physiol, 558: 479-488, 2004

    Koyama et al, J Physiol, 532: 759-769, 2001

    Woodward et al, Am J Physiol, 297: L954-L964, 2009

        The PI3K/Rho/ROCK signalling pathway is now mentioned at lines 73-75. Refs byWoodward et al            and Koyama et al have ben added (n . 31, 32).

  • In addition, the reference to purinergic regulation of intracellular metabolism should be included in Part 4 as it supports the idea of purinergic regulation of cancer metabolism.  

    Lapel et al, Am J Physiol, 312:C56-C70, 2017.

        Reference to Lapel et al has been added (n 135), an an introductory sentence has been added to            the text (lines 312-313).

  • 4. Figure 2 size should be scaled down to fit the figure legend to the same page.

        We reduced the size of Figure 2, but were still unable to fit the legend in a single page. 

  • 3. A summary of clinical trial and/or ATP-based anti-cancer drugs would be helpful to appreciate the translational aspect of the review. 

        A new Table (Table I) is now shown in the review. This Table summarizes clinical trials aimed at                modifying the eATP concentration acting at CD39 or CD73, or that exploit the increased eATP                levels of the TME for therapeutic purposes. See also lines 400-402.